



# Mapping snow depth and volume at the alpine watershed scale from aerial imagery using Structure from Motion

Joachim Meyer[1], S. McKenzie Skiles[1], Jeffrey Deems[2], Kat Bormann[3], David Shean[4]

[1] Department of Geography, University of Utah, Salt Lake City, UT, USA
[2] National Snow and Ice Data Center, Boulder, CO, USA
[3] Airborne Snow Observatories, Inc., Mammoth Lakes, CA, USA
[4] Dept. of Civil and Environmental Engineering, University of Washington, Seattle, WA, USA

*Correspondence to*: Joachim Meyer (j.meyer@utah.edu)

## Abstract

Time series mapping of water held as snow in the mountains at global scales is an unsolved challenge to date. In a few locations, lidar-based airborne campaigns have been used to provide valuable data sets that capture snow distribution in near real-time over multiple seasons. Here, an alternative method is presented to map snow depth and quantify snow volume using aerial images and Structure from Motion (SfM) photogrammetry over an alpine watershed (300 km$^2$). The results were compared at multiple resolutions to the lidar-derived snow depth measurements from the Airborne Snow Observatory (ASO), collected
simultaneously. Where snow was mapped by both ASO and SfM, the depths compared well, with a mean difference between -0.02 m and 0.03 m, NMAD of 0.22 m, and close snow volume agreement (+/- 5%).  ASO mapped a larger snow area relative to SfM, with SfM missing ~14% of total snow volume as a result. Analyzing the differences shows that challenges for SfM photogrammetry remain in vegetated areas, over shallow snow (< 1 m), and slope angles over 50 degrees. Our results indicate that capturing large scale snow depth and volume with airborne images and photogrammetry could be an additional viable
resource for understanding and monitoring snow water resources in certain environments.



## 1. Introduction

Snow depth and snow water equivalent are essential monitored quantities and applied to many water resource applications. In alpine environments, snow depth is traditionally measured continuously at instrumented sites or periodically along transects due to the complexity of the terrain. These long-term records are valuable but tend to be located at sites that are accessible,
lower in elevation, and hold snow for longer than the surrounding terrain (Molotch and Bales, 2005). This limited spatial coverage leaves a poor understanding of snow depth distributions in the mountains, particularly at high elevations. The gap can be addressed by mapping snow depth differentially using remotely sensed surface elevation products (Deems et al 2013). Snow depth can be estimated with a pixel-wise calculation on raster-based products that subtracts snow-free elevations from snow-on elevations over a target area.

This principle has been demonstrated from several remote sensing platforms, spanning a range of spatial resolution, areal coverage and repeat intervals which resulted in varying degrees for monitoring snow depth accuracy. The Ice, Cloud, and Land Elevation Satellite (ICESat) was a laser altimeter that could map snow depths along swaths with sub-meter accuracy (Treichler & Kääb, 2017), but the data are of limited utility for mountain regions due to the low temporal resolution and large ground footprint (70 m). Satellite stereo photogrammetry-derived DEMs, such as those from WorldView or Pléiades, have the potential
for higher spatial (< 1 m) and temporal resolutions (Shean et al., 2016, McGrath et al., 2019, Deschamps-Berger et al., 2020). Limitations, though, include reduced accuracy in complex terrain, 10–50 cm over shallow slopes (<10°; Shean et al., 2016), and requiring clear view to the target area with no obstruction such as clouds for example (Shaw et al., 2020). With no current space-borne platform providing the combination of high temporal and spatial resolution required for accurate distributed snow mapping, airborne campaigns have been established to address these needs. For example, the Airborne Snow Observatory
(ASO), combining a Light Detection and Ranging (lidar) and imaging spectrometer platform, delivers time-series of snow depth maps at 3 m resolution with centimeter accuracy in select watersheds primarily in the California Sierra Nevada and Colorado Rocky Mountains (Painter et al., 2016). Although smaller in spatial extent and periodic, it has been shown that Structure from Motion (SfM) photogrammetry using imagery from Remotely Piloted Aircraft System (RPAS) can map snow depth at sub-decimeter resolution, while maintaining centimeter accuracy for areas up to alpine catchments size (< 1km$^2$)
(Bühler et al., 2016; Harder et al., 2016; Schirmer & Pomeroy, 2020).

Airborne platforms have important limitations, such as expense and logistics for a piloted aircraft, weather restrictions for remotely piloted and piloted aircraft, and limited areal coverage with RPASs due to battery life as well as the ability to operate safely in line of sight to the RPAS. Limitations lead to significantly smaller footprints and less consistent coverage relative to space-borne platforms. Still, the ability to acquire high resolution/high accuracy data sets on-demand, over any desired target
area, makes airborne campaigns an essential tool for both water management operations and research in alpine environments. The resulting data sets have expanded our knowledge of snow science in watersheds (Behrangi et al., 2018; Brandt et al., 2020; Hedrick et al., 2018, Zheng et al., 2019) and are now well established as relevant data sources. In particular, the RPAS-SfM





studies have seen a recent gain in popularity due to high affordability using consumer-grade cameras that can deliver highly accurate data sets (Gaffey & Bhardwaj, 2020).

This paper evaluates the ability of SfM to map snow depth distributions over a watershed with high-resolution imagery captured by a piloted aircraft relative to coincidentally collected lidar-based retrievals. The existing literature comparing photogrammetric snow depth mapping from high altitude piloted aircraft (Bühler et al., 2015; Nolan et al., 2015; Eberhard et al., 2020) have been either compared to manual measurements, had differences in the time of day for the recording, or did not evaluate the complete extent of the study area. Additionally, the coincidental collection with this study enables a comparison

with identical viewing perspectives and environmental influences, such as the weather, as well as precluding measurement errors due to manual recording such as snow probes. The used data set is also sourced from a regular repeated observation, which could readily adapt this presented workflow into their daily operations. *Meyer & Skiles* (2019) showed that accurate DEMs can be generated from imagery collected from piloted aircraft over bright snow surfaces using SfM. Building upon this work, we show that SfM DEMs can be used to differentially calculate snow depths and corresponding snow volume over a

relatively large alpine watershed (300 km$^2$) at scales commensurate with airborne lidar-based applications. This comparison demonstrates that SfM is a reliable remote sensing technique for large-scale DEM reconstruction and differential volume mapping in complex terrain, and will further expand our understanding of the strengths and weaknesses of applying photogrammetric-based techniques to automate areal snow observations.

**2. Study area**

The East River watershed (330050 E E,4314650 N; EPSG:32613), located northeast of Crested Butte, Colorado, lies within the broader Upper Gunnison watershed. It encompasses the long running Rocky Mountain Biological Laboratory and a

portion of Crested Butte Mountain Resort. The East River is one of two primary tributaries of the Gunnison River, which itself discharges into the Colorado River. The watershed is an estimated 300 km$^2$ in size and has an average elevation of 3266 m and vertical relief of 1420 m (Hubbard et al., 2018). The

vegetation varies across the elevation ranges and includes brush and grass land, aspen and mixed conifer, and alpine meadows. The East River was designated as a Scientific Focus Area in 2016, supported by the US-DOE Biological and Environmental Research Subsurface Biogeochemistry Program. The Airborne

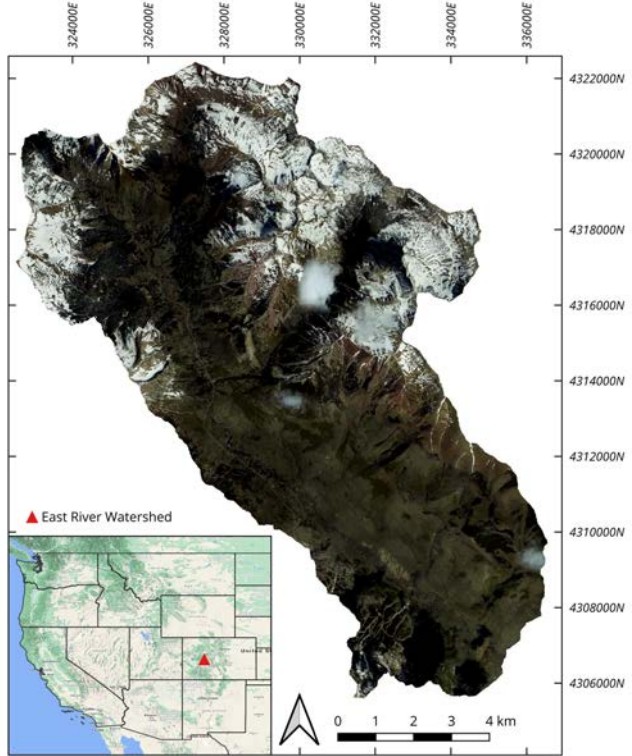

**Figure 1 - Areal overview of the East River watershed and it's location shown relative to the Western United States. (Insert map base layer: Map data ©2021 Google)**



Snow Observatory flights, and subsequent data processing, were funded by the state of Colorado to map snow distribution patterns and support water supply forecast improvements.

## 3. Data

The ASO flights took place on the 24$^{th}$ May 2018 for the snow-on scene and 12$^{th}$ September 2019 for the snow-free. Flight patterns were almost identical on both days, covering the area with a 50% overlap lawn-mower pattern. Flight altitude varied

slightly between the two flights, where the May flight was 6400 m above sea level, and the September flight was at 6100 m. There was also a difference in flight line orientation between the two dates, with the May flights in a North-South direction and the September flights in a Northwest -Southeast direction. The orientation for the flight lines during the snow season were selected based on lighting conditions for the ASO imaging spectrometer and flight efficiency, and no direct considerations for the camera are given.

The camera used by ASO is mounted inside the lidar instrument, which creates identical view perspectives between the lidar scanner and the camera to the ground surface. Each image has dimensions of 10,328 × 7,760 pixels with a 16-bit color depth and size of 5.2 micron for an individual pixel. Underlying hardware consisted of a medium format Phase One iXU 180-R CCD sensor camera with a Rodenstock 50 mm HR Digaron-W wide-angle view lens. The recording interval for the camera was twelve seconds for the snow-free flight, which resulted in 287 images, and six seconds for the snow-on flight resulting in 582

images. The average ground sample distance (GSD) was 0.31 m/pixel for the snow-on and 0.28 m/pixel for the snow-free images. An overview for both collections is shown in Table 1.

For quality assessment of the measured depth by SfM, we used the publicly available snow depth product by ASO, which is published through the National Snow and Ice Data Center. ASO uses the identical difference principle to calculate depth, where snow-on values are subtracted with the snow-free. More technical details on the ASO platform and the final output

product's processing steps can be found in Painter et al., 2016.

**Table 1 - Flight parameters for snow-on and snow-free recording.**

|  | *24 May (snow-on)* | *12 September (snow-free)* |
|---|---|---|
| Flight pattern | Single overlap, lawn-mower | Single overlap, lawn-mower |
| Flight line orientation | North-South | Northwest-Southeast |
| Flight altitude (above sea level) | 6400 m | 6100 m |
| Camera recording interval | 12s | 6s |
| Number of images | 287 | 582 |
| Mean GSD | 0.31 m/pixel | 0.28 m/pixel |





## 4. Methods

### 4.1 Image Processing

The camera images from the ASO survey were processed using Agisoft Metashape (version 1.6.2) along with associated geo-location and orientation data from the airplane global navigation satellite system and inertial measurement unit. Metashape
was used for feature matching, image alignment, and dense point cloud creation. We refer readers for more technical details on data preparation and settings for Metashape to the workflow in *Meyer & Skiles* (2019).

### 4.2 Co-Registration

After the SfM snow-free and snow-on point clouds were generated, a reference lidar elevation data set ensured the closest alignment of surface models through co-registration. This process minimizes relative geo-location error and provides improved
accuracy for DEM difference products. The co-registration was performed using the Ames Stereo Pipeline (ASP; version 2.6.3), which internally uses the iterative closest point algorithm to determine the difference between a reference and movable point cloud (Shean et al., 2016, Beyer et al., 2018). As a reference, the ASO snow-on acquisition flight was used, which data consisted of control surfaces. These surfaces were identified from the ASO imaging spectrometer classification and are consistent elevation across time, such as exposed bedrock or roads. The control surfaces were additionally refined by removing
any areas that had snow in the ASO snow depth product and any slopes steeper than 50 degrees (Shaw et al., 2020). The bounding box for the reference DEM was extended beyond the watershed boundaries to increase the available area for co-registration (Figure 2b). An added advantage of co-registering of the SfM point clouds to the ASO lidar point cloud was the implicit alignment with the ASO snow depth product.

The co-registered point clouds were converted to a gridded raster product (GeoTIFF) with 1 m resolution using the Point Data
Abstraction Library (PDAL, Contributors, 2018), which provides the inverse distance weighting (IDW) algorithm for interpolation. The IDW algorithm can be applied with a point density of multiple points per square meter (Guo et al., 2010), and both SfM clouds had sufficient density for its application at the 1 m resolution (23.2 points/m$^2$ for the snow-on and 31.5 points/m$^2$ for the snow-free). In addition to the resolution and algorithm, PDAL was also used to clip the outputs to identical bounding boxes and transform to matching projection (WGS 84 / UTM zone 13N; EPSG 32613). The final step was calculating
the SfM snow depth by taking the pixel wise difference in surface elevation between the snow-on and snow-free DEMs.

### 4.3 Comparison

The snow depth (SD) values from SfM were compared to the ASO snow depth map by treating ASO as the reference, since snow depth mapping with lidar is the more established method. ASO distributes its snow depth products at 3 and 50 m resolution (Painter et al., 2016, Painter, 2018a, b) and both were used in this comparison. We additionally compared the
products at the 1 m resolution with SfM supporting the higher resolution through the previously mentioned high point density, and down sampled the 3 m ASO snow depths for this iteration. The 1 m resolution also enabled a higher level of detail when



investigating possible influencing terrain factors such as slope, as well as better retrieval of snow depth values around forested areas with SfM.

The SfM snow depths for all resolutions were compared to ASO snow depths using the mean, median, and standard deviation
for the full study domain. Additionally, we calculated the snow-covered area (SCA), snow volume, and the resulting SWE. For consistency, the SWE calculation used the mean snow density calculated from the ASO SWE product distributed through NSIDC at the 50 m resolution (Painter, 2018c). The analysis for terrain factors was based on the 1 m resolution product and first binned the depths by elevation to assess similarities in the vertical relief. Next, a relative pixel-by-pixel difference comparison between the two data sets, calculated by subtracting the $SD_{SfM}$ from $SD_{ASO}$, included the mean, median, standard
deviation, and normalized median absolute deviation (NMAD; Höhle and Höhle, 2009). Finally, the snow volume was compared, grouped by different surface classifications (snow, rock, vegetation; Figure 2a). We note that the water class from the imaging spectrometer was mostly misclassified, it was shading within vegetation, which we confirmed with a subset of all water classified pixels. Therefore, we treated both categories (water and vegetated areas) as one category. Additionally, scaling the ASO snow depth map from 3 to 1 m, combined with the fact that the ASO imaging spectrometer is not processed in a way

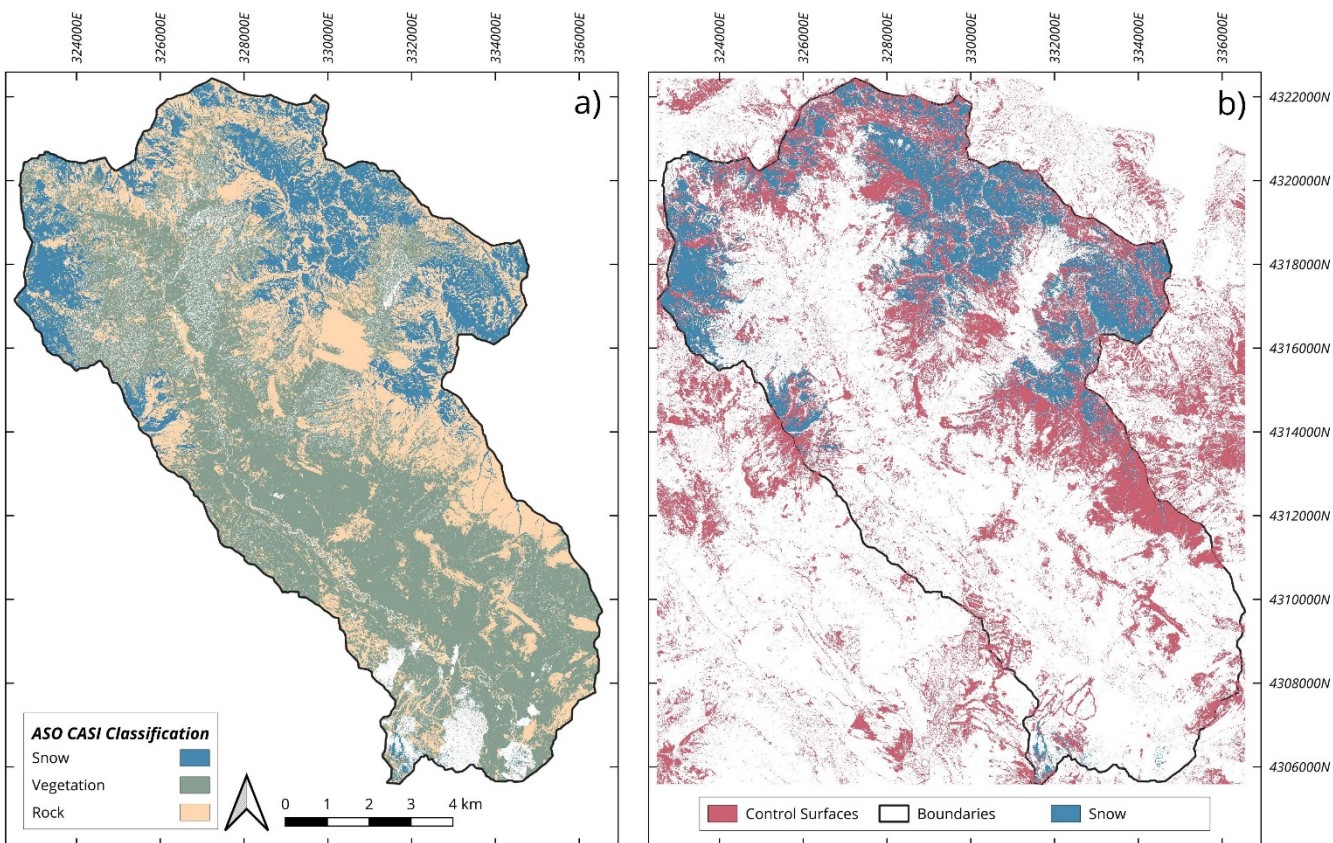

**Figure 2 – ASO imaging spectrometer classification of the study area. Most snow surfaces were in the higher elevations in the northern locations (a). Control surface (red) distribution over the extended watershed boundaries used for co-registration (b).**





to map fractional land cover, resulted in measured snow depth in pixels classified as rock in the snow-on data. Overall, the focus was to check for similarity in distribution pattern and volumetric agreement for SfM relative to ASO.

Negative SfM snow depth values, treated as SfM reconstruction and/or co-registration errors, were inspected by terrain characteristics (elevation, slope, and aspect) for the full domain, and by surface classification type from the imaging spectrometer. Aspect and slope were calculated from snow-free lidar acquisition by ASO to create independence from the

modeled values by SfM. Median and NMAD for stable terrain differences, using overlapping areas between the SfM snow-free and snow-on DEM, determined the relative error of the two models.

## 5. Results

### 5.1 Co-registration

Control surfaces, used for co-registration of the SfM snow-free and snow-on scene to the lidar reference point cloud,

encompassed 13.9% of the watershed boundaries when gridded at the 1 m resolution (Figure 2b). The snow-free point cloud was shifted 0.02 m to the North, -0.20 m to the East and -0.41 m in the vertical direction, while the snow-on was 0.01 m to the North, -0.02 m to the East and 0.01 m in the vertical. After applying the translation, the differences over the control surfaces in the raster products exported from the respective SfM point clouds had a mean of 0.02 m with a standard deviation of 0.52 m, and median of 0.03 m, with a NMAD of 0.22 m. The remaining difference in the NMAD indicated that there were still

some outliers in the control surfaces, despite all the refinements to constrain those. With median and mean close to zero, however, the co-registration can be considered successful for the two scenes. The NMAD can also be used as a measure for uncertainty in the snow depth values calculated from the two SfM DEMs.

### 5.2 Structure from Motion snow depth

Overall, where $SD_{SfM}$ and $SD_{ASO}$ were measured in both the SfM and ASO depth maps, hereafter referred to as 'SfM' and

'ASO' respectively, there was good agreement in both snow depth and snow volume (Figure 3). Notably, agreement was best for deeper snow (> 1 m) and across higher elevations. There were few gaps, indicating the image sampling configuration was of high enough quality, and with sufficient overlap, to provide a reliable source for reconstruction by SfM. Most of the unsuccessful SfM measurements were in areas with vegetation or shallow snow depth and the total area varied across the different output resolutions (Figure 4).





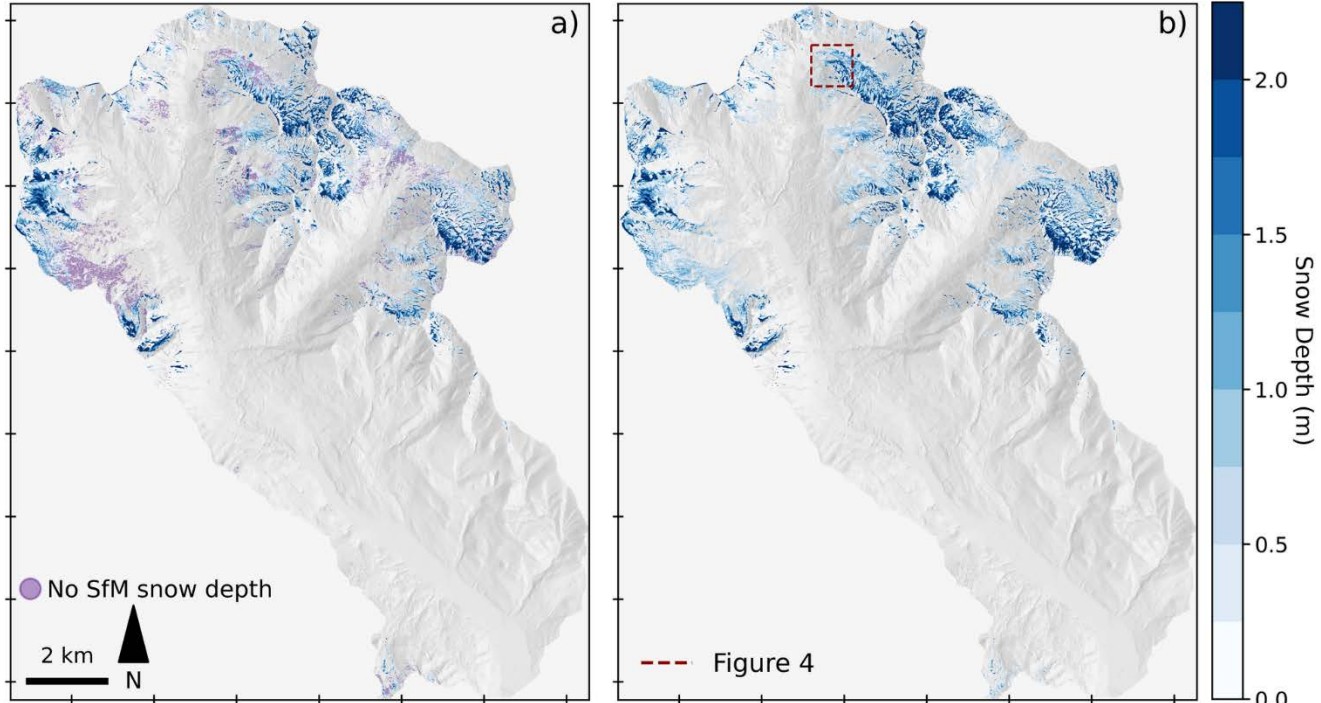

**Figure 3 - Overview of reconstructed snow depth by SfM (a), with ASO snow depth map shown on the right (b) at 1 m resolution. Areas with unsuccessful SfM measurements (purple) coincided with surfaces classified as vegetation or shallow snow depth values (< 1 m) by ASO. The snow depth pattern between the two products show good agreement over the overlapping area. The red dashed box shows the location of the comparison in Figure 4.**

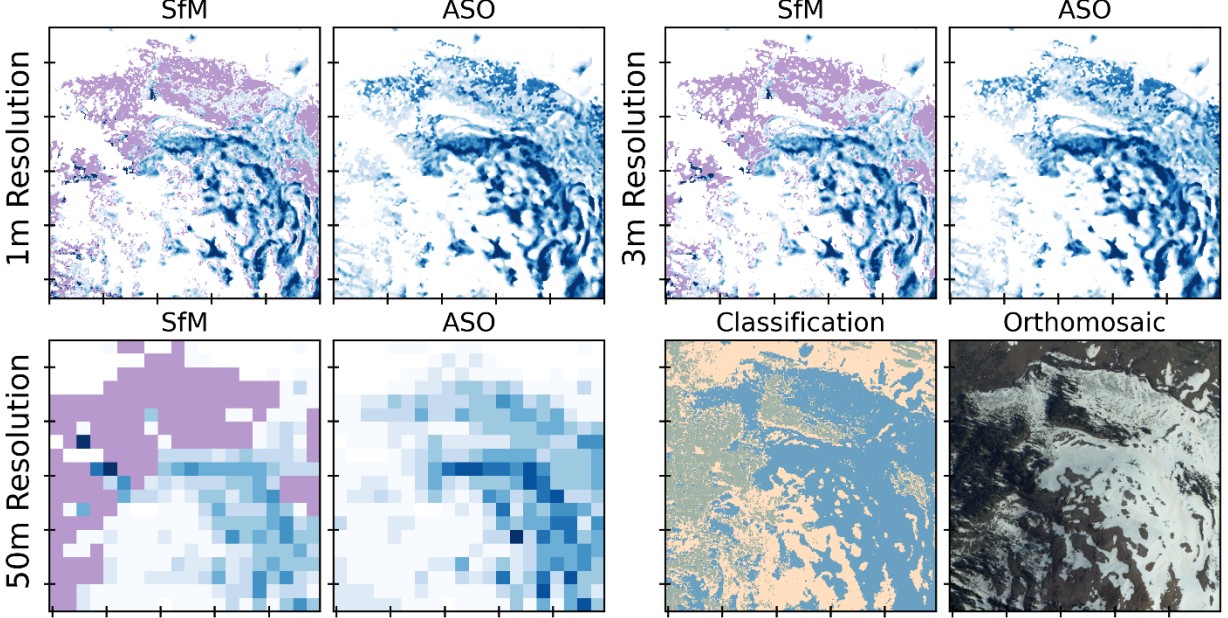

**Figure 4 – Comparison of snow depth across 1 m, 3 m, and 50 m resolution. SfM did not accurately capture snow depth within vegetated areas, but showed good agreement for open areas. Higher resolutions at 1 m and 3 m were able to compensate for missing areas at 50m and successfully measured more areas with snow depths at edges of vegetation were more distinct.**





### 5.2.1 Snow Depth, Snow Volume, SWE, and SCA

Mean snow depth measured by ASO in the distributed product through NSIDC was 0.89 m, with a median value of 0.64 m, and standard deviation of 0.88 m for the entire domain at the 3 m resolution, and 0.60 m (Mean), 0.40 m (Median), and 0.66 m (Standard Deviation) at 50 m. The snow depths statistics for SfM, ASO, and the pixel-wise difference for the overlapping

area are shown in Table 2. All resolutions showed almost identical values for the mean and median, while the standard deviation had the biggest difference. The higher standard deviation in the pixel-wise difference is attributed to outliers caused by mixed pixels in SfM that had vegetation and snow cover within one pixel. The SfM SCA coincided with ASO at the 1 m resolution by 72%, 73% at 3 m, and 64% at 50 m, showing a small difference between the lower and higher resolution.

The snow volume had a close match where snow was mapped by both ASO and SfM, with SfM having 1% higher total volume

(101% compared to ASO) or $21.10 \times 10^6$ m$^3$ at the 1 m resolution. The SfM volume was 86% of the ASO measured snow volume for the entire watershed, which translated to a difference of $3.42 \times 10^6$ m$^3$ to the ASO total of $24.52 \times 10^6$ m$^3$. Similar match for the volume was attained at the 3 m resolution, showing a slight underestimation of 2% in the overlapping area (98% compared to ASO) with a total volume of $20.63 \times 10^6$ m$^3$. This resulted in an 84% match of the measured volume for the entire watershed or a negative difference of $3.90 \times 10^6$ m$^3$. SfM snow volume had a slight overestimation at the 50 m resolution,

reporting a 5% higher value compared to the overlapping ASO area at $21.12 \times 10^6$ m$^3$. The resulting discrepancy to the total watershed volume was $3.31 \times 10^6$ m$^3$ or a match of 87%. Overall, the difference in overlapping area was consistent across the different resolutions, signifying that SfM can support scaling to a similar range of image resolutions compared to lidar.

The estimated amount of SWE was calculated by applying a constant snow density of 394 kg/m$^3$, the median value calculated from the 50 m ASO SWE/depths maps. This is a simple treatment of snow density, but in the spring snow density varies less

**Table 2 - Overview of snow depth statistics across the three different resolutions.**

| **1 m** Resolution Snow Depth | **SfM** | **ASO** | *Difference (ASO-SfM)* |
|---|---|---|---|
| Mean Depth (m) | 1.06 | 1.05 | 0.01 |
| Median Depth (m) | 0.76 | 0.79 | -0.03 |
| Standard Deviation (m) | 1.11 | 0.96 | 0.83 |
| **3 m** Resolution Snow Depth | | | |
| Mean Depth (m) | 1.03 | 1.05 | -0.02 |
| Median Depth (m) | 0.74 | 0.79 | -0.04 |
| Standard Deviation (m) | 1.06 | 0.96 | 0.75 |
| **50 m** Resolution Snow Depth | | | |
| Mean Depth (m) | 0.62 | 0.60 | 0.03 |
| Median Depth (m) | 0.41 | 0.40 | -0.01 |
| Standard Deviation (m) | 0.76 | 0.86 | 0.67 |

*Note: Mean, Median, and Standard Deviation are for overlapping area by SfM.*





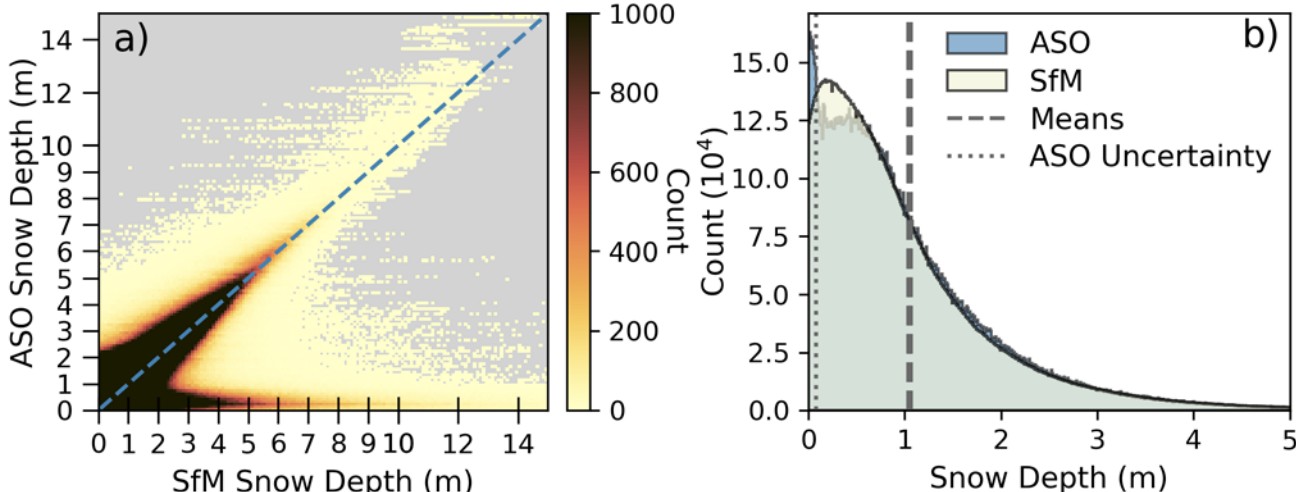

**Figure 5 -** SfM snow depth values plotted against ASO snow depth values at 1 m resolution (a). The dashed line shows a hypothetical one-to-one relationship. SfM tended to underestimate the snow depth compared to ASO. The snow depth histograms showed a strong agreement between the two sources (b).

than snow depth. With this single value is applied to the ASO snow depth map, the calculation of basin SWE matched the official ASO product (3770 m total SWE) within 3%. Based on this estimate of SWE, SfM captured 86% of ASO mapped SWE at the 1 m, 84% at the 3 m, and 87% at 50 m resolution, which resulted in missed SWE of 526 m (1 m), 599 m (3 m), and 508 (50 m).

As a whole, SfM showed an underestimation of snow depth compared to ASO, using the 1 m resolution pixel-by-pixel values

(Figure 5a). The depth distribution was mostly in the 0 to 5 m range (Figure 5b), and higher values were more dispersed and highly localized. Other studies that have used ASO snow depth maps also observed extreme outliers and considered these as spurious snow depth values (5 m in McGrath et al., 2019; 6 m in Brandt et al., 2020). For this study, we did not remove any high outliers from both data sources and included them in all comparisons.

### 5.2.2 Influence of Terrain

A closer look at the coinciding area at the 1 m resolution had most SfM land surface pixels classified as open snow (81%), followed by rock (12%), and then vegetation (7%). In these land surface categories, SfM underestimated snow volume relative to ASO in the pixels classified as snow, capturing 92%, while overestimating snow volume in the rock and vegetation pixels. For ASO, snow volume was distributed differently across land surface types in the entire watershed; 69% in open snow, 15% in rock, and 16% in vegetation. This indicates, in part, that SfM is less likely to successfully map snow in vegetation.

The distribution of snow depth values across 10 m elevation bands showed higher accumulation in the upper elevations, but not necessarily at the highest elevations in both data sets (Figure 6). Both SfM and ASO had increasing depths between ~3200 m and ~ 3800 m, with depths started to decrease above this range. The values from SfM, however, had a higher spread in the elevations between 3200 m and 3500 m (Figure 6a), with 74% of this elevation band classified as vegetation. This greater



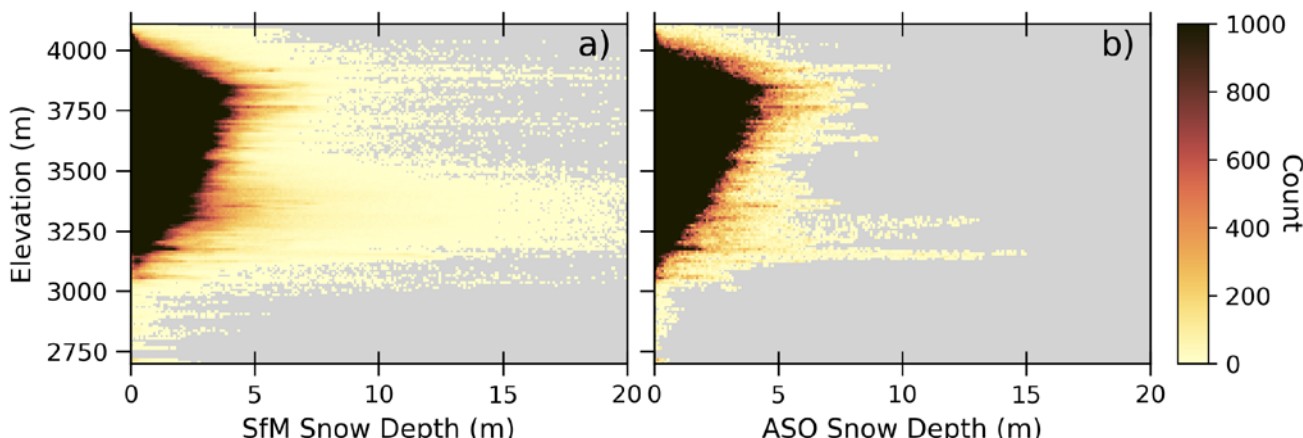

**Figure 6 - Snow depth distribution across elevation bands of 10 m for SfM (a) and ASO (b) at 1 m resolution. Both had similar patterns of higher depth values in the upper elevations and between 3600 m and 4000 m. The higher snow depth spread in the lower elevation between 3200 m and 3500 m by SfM is attributed to more areas with vegetation.**

noise pattern was expected, as vegetated areas remain a challenging environment for SfM to measure snow depth (Harder et

al., 2020). As a whole, the agreement between the two distributions shows that SfM can be used to map snow depth patterns

across a range of elevations in complex terrain.

### 5.3 Structure from Motion measurement errors

The area mapped as snow by ASO, but not by SfM, spanned 28% of the snow-on area present in the ASO depth map at the 1

m resolution, of which 0.5% was a data gap in SfM. Gaps had no measured snow, while the 'missed' area returned negative

snow depth values. The snow depth that was missed by SfM had a mean of 0.48 m, median of 0.39 m, and a standard deviation

of 0.42 m.

To further investigate the pixels with no measured depth from SfM, we compared them to the corresponding snow depth values

from ASO (Figure 7). This showed that SfM failed to map snow where ASO mapped shallower snow (<1 m; Figure 7a) in

open areas, and the largest negative SfM values (-5 m and -28 m) were primarily found in areas classified as vegetation (Figure

7b). As previously mentioned, vegetation is challenging for SfM and these results are in line with previous SfM work, where

shallow depth or forested areas were shown to impair the ability of SfM to measure accurate surface elevations (Avanzi et al.,

2018; Fernandes et al., 2018).





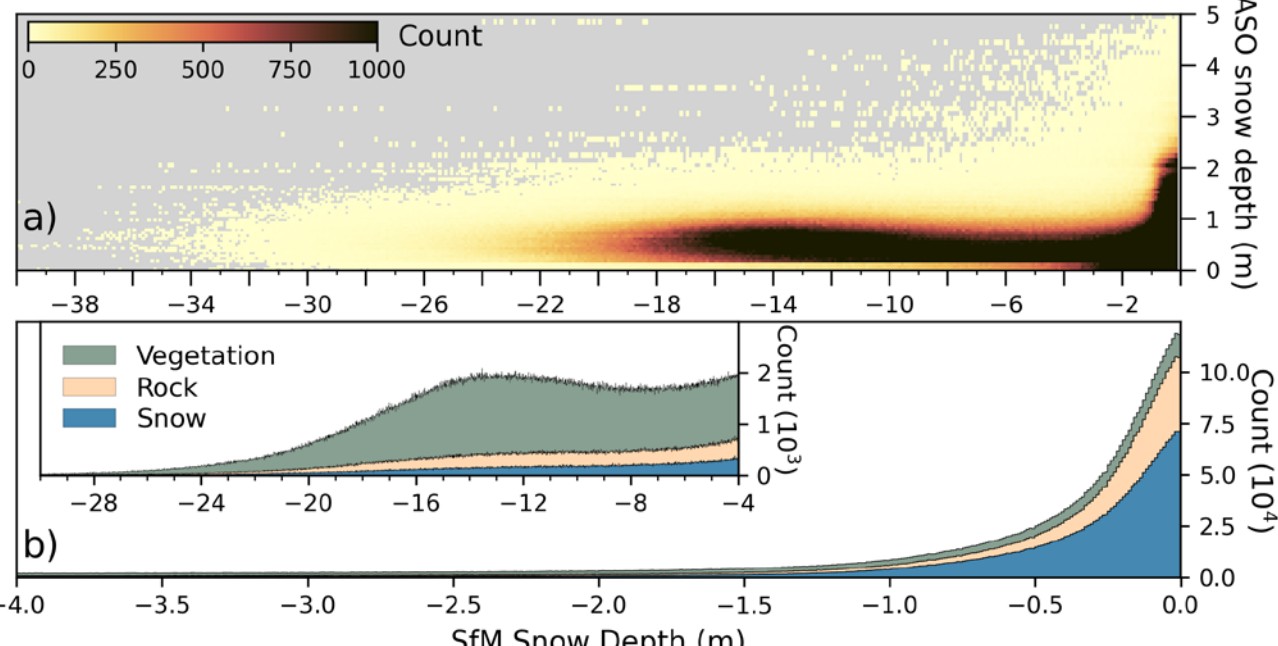

**Figure 7 - Snow depth with negative values by SfM plotted against values by ASO at 1 m resolution. Extreme outliers are dominantly found in areas with snow depth of less than 1 m (a) or vegetated areas (b).**

An analysis to correlate areas of snow missed by SfM with terrain characteristics showed no strong relationships for the area as a whole, or across land surface classification types. Out of the investigated influence factors of aspect, elevation, and slope,

the most visible trend was detected when values filtered to only open areas (no vegetation) were binned by slope angle and the median depth calculated. The median did not exceed -1 m and showed a linear trend up until around 55 degrees, then started to decrease sharply (Figure 8). This observation is similar to other studies, where slopes above 50 degrees show a decline in accuracy for photogrammetric reconstructions (Shean et al., 2016; Shaw et al., 2020).

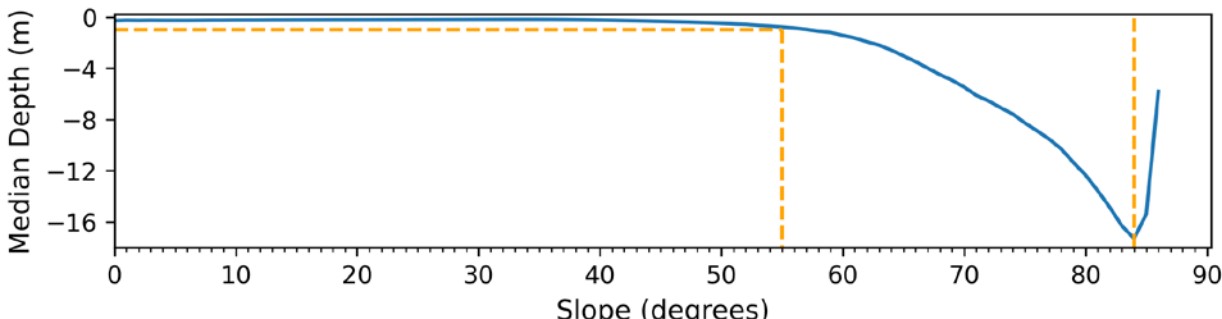

**Figure 8 – Median SfM snow depth in open areas binned by slope angles showed a linear trend until 55 degrees and stayed below -1 m (orange dotted lines), before increasing sharply. This trend was the only strong relationship detected when analyzing the negative SfM values for possible influences due to terrain.**



## 6 Discussion

### 6.1 Structure from Motion with airplane imagery


The primary focus of ASO is the delivery of lidar-based snow depth and snow water equivalent maps, and the camera images are not currently used as a resource in data product processing (Painter et al., 2016). With the lidar and imaging spectrometer as the primary data streams, there is little consideration given to the image overlap, illumination conditions for the camera sensor, or minimum GSD for further use with photogrammetric reconstruction. Given this image acquisition setup and the

presented results, we believe that although the results here are promising, there is room for improvement if flight campaigns were planned to produce snow depth maps with SfM. For instance, consistent image overlap can improve the quality of SfM output products (Bühler et al., 2016; Harder et al., 2016; Meyer and Skiles, 2019). The potential for snow-depth mapping by SfM has been demonstrated on a smaller scale by Nolan, et al. (2015) with an accuracy of +/- 0.3 m. Our NMAD of 0.22 m over a larger target area denotes the scalability of this technical setup and is in line with Eberhard, et al. (2020), where the

NMAD was 0.17 m. Another indicator for the capability of SfM is shown by the point density of the two SfM point clouds. Here, we had an average of 23.2 points/m$^2$ for the snow-on acquisition and 31.5 points/m$^2$, which signifies well re-constructed surfaces by SfM. With the high point density, higher resolutions for the gridded output products are possible, as well.

We acknowledge that the combination of a reference lidar and comparison snow depth data with a single acquisition is unique to ASO's recording setup. The data set provided an opportunity to perform a comparison of SfM to an established snow depth

mapping technology, but we note that it is not needed to perform this methodology outside of the ASO operation domains. For classification, the snow-on and snow-free point clouds can be directly classified using the SfM point clouds and the image RGB information (Shaw et al., 2020) or near-infrared spectrum (Deschamps-Berger et al., 2020), where available. Producing the classification with this approach was beyond the scope of this work and warrants an accuracy assessment by itself before continuing to use in downstream products. Using the existing classification, we reduced a potential source of error for the

depth and volume assessment. Once classified, ensuring proper geo-location of the models can be completed by co-registration against suitable control surfaces from any externally sourced point or gridded based referenced data set and solutions to complete this already exist (Shean et al., 2016). For areas with little change to control surfaces (exposed rock surfaces or roadways), the reference DEM can further be from different recording years and does not have to be acquired within the same year of the images (Midgley & Tonkin, 2017). In the end, the presented processing steps can be applied to any airborne

collected and geo-referenced image data set. A lidar-based reference or image spectrometer classification is not required.

### 6.2 Absence of Ground Control Points

Ground control points (GCP) are commonly used for RPAS based studies to geo-reference their results, which strongly influences accurate geo-location (James et al., 2017). Our process explicitly precluded the use of GCPs to reduce manual processing intervention and increase automation potential. We believe that image geo-location and perspective information

combined with co-registration is a reliable substitute to GCP's, while not compromising on output quality. Co-registration is



a common practice for photogrammetric snow depth products from satellite image (Shean et al., 2016; McGrath et al., 2019; Shaw et al., 2020; Deschamps-Berger et al., 2020) and equally applicable for areal imagery across larger alpine areas. The SfM software performed very well with the image metadata and the snow-on model was very close to the lidar, while the snow-free model had a higher shift, predominantly in Y (-0.20 m) and Z (0.41 m) direction. We hypothesize that the snow-

free scene, with more exposed vegetation and ground cover, degraded the accuracy for SfM. With both alignment adjustments very low in magnitude, it is further feasible to align the two models to each other and compute snow depth and volume in relative geo-location space. Alpine areas benefit from having exposed control surfaces for multi-view image processing and co-registration, having identifiable features in both scenes. For different environmental conditions, such as ice-sheets that have little to no overlapping stable terrain, alternative approaches have been developed to align corresponding surfaces (Howat et

al., 2019; Shean et al., 2019).

### 6.3 Influence of image resolution

The similar results for depth across the different image resolutions of 1 m, 3 m, and 50 m demonstrates that SfM has the capabilities for accurate retrieval compared to a lidar based platform on large scale. The overlapping pixel wise difference for the mean and median snow depth across the resolutions varied between -0.04 m and 0.03 m and falls within the reported

accuracy for ASO snow depth maps at the 3 m resolution of 0.05 m (ASO, personal communication). The standard deviation had a bigger difference (0.83 m to 0.67 m), where we see pixels with vegetation contributing the most to this higher value. The lower end of 0.67 m at 50 m resolution is also attributed to the reduced error in the snow depth values at coarser resolutions and has been previously demonstrated with other photogrammetric based platforms (Deschamps-Berger et al., 2020). We argue that the decisions on the chosen image resolution also comes down to the use case of the snow depth map. Inputs for hydrologic

models, for instance, currently do not require or have the capabilities to use image resolutions of less than 25 m (Behrangi et al., 2018; Hedrick et al., 2020; Pflug and Lundquist, 2020). The presented results for SWE, and difference between the 1 m and 50 m there, also show that there is not an imminent need to calculate based on a high-resolution product (86.05% at 1 m versus 86.52% at 50 m). For those applications we see a strong use case for SfM based retrievals as our results showed good agreement across metrics of depth and volume at the lower resolutions.

### 290  6.4 Comparison to other platforms

The SfM NMAD from airplane imagery in this study shows a higher accuracy compared to satellite-based stereo photogrammetric studies, where the NMAD ranges from 0.36 m (Shaw et al. 2020) over 0.45 m (Marti et al. 2016) and up to 0.69 m (Deschamps-Berger et al., 2020). Reasons for the higher accuracy can be topographical, as satellite stereo pairs have a larger area with more varying terrain in a single scene, which makes it more difficult to capture high enough detail of

information for reconstruction. Additionally, DEM generation from satellite images has a different technical setup, where stereo photogrammetry uses up to three images (tri-stereo) (Shaw et al, 2020; Deschamps-Berger et al.; 2020, Bhushan et al., 2021). This varies for SfM, which can use any number of images, driven by the amount of overlap in an area. Weather



conditions are an additional high impact factor when using satellite imagery. An unobstructed view from an instrument to the entire study area at the time of overpass cannot be guaranteed, and atmospheric features like clouds can cause additional occlusions. On the smaller scale using RPAS platforms, the accuracy is higher (cm scale) compared to what we have achieved here (Avanzi et al., 2018, Harder et al., 2016, Bühler et al., 2016). This can be attributed to the lower flight altitude and resulting higher degree of image overlap and GSD. One of the remaining challenges for RPASs, though, is the ability to cover larger areas with limited battery life, higher sensitivity to weather conditions in alpine areas, and access challenges to operate safely (Bühler et al, 2016).

Given these limitations, we see piloted aircraft SfM filling an important gap between RPAS-SfM and satellite stereo photogrammetry. Across the different SfM output resolutions and the accomplished NMAD here, we believe that SfM compares well against an active measurement instrument like lidar on larger scales. This assessment holds for open spaces with a snowpack of more than 1 m. As with other studies, we see vegetated areas and shallower depth a remaining challenge for SfM, and the technology still needs to be improved in order to match results that are possible with lidar (Harder et al., 2020). Steep terrain is another aspect where accuracy for the results degrades, particularly on angles above 50 degrees. Here, we argue that the accumulation in those areas is low and also see a need in the literature to assess the quality of remote sensing measurements.

**6.5 Expanding snow science**

The ability to fill missing information between point-based snow depth measurement locations has improved our understanding of large-scale snow processes. Data sets from airborne campaigns have been used to improve model capabilities to predict snow precipitation (Behrangi et al., 2018), observe snowfall distributions (Brandt et al., 2020), or improve snow energy balance models (Hedrick et al., 2020). Expanding the number of observed regions with spatially and temporally extensive records can further accelerate our ability to understand snow processes at scale and improve estimates for SWE (Margulis et al., 2019) or streamflow. (Shaw et al., 2020) Although SfM is not yet able to deliver similar accuracies to lidar for all terrain characteristics and land cover classes, it can be used to supplement or build upon lidar data sets. For instance, a first survey could be conducted using the more accurate lidar, and successive observations use the more cost-efficient SfM for open areas with little forested areas (Pflug and Lundquist, 2020). With the results of this work, we are encouraged that SfM can be an option for operations like ASO for repeated observations after the initial lidar flight. From a technical setup perspective, it is further feasible to source the images from space-borne platforms, adding the option of temporally consistent broad-scale coverage and reduce operational requirements. This perspective is also a potential for additional research, where more cross-platform (airborne vs spaceborne) and technology (photogrammetry vs lidar) comparisons like Eberhard et al., 2020 are needed to fully evaluate our capabilities on hand.



## 7. Conclusions

This study's motivation was to investigate whether Structure from Motion should be considered an additional remote sensing data source for snow depth monitoring on a large watershed scale. It also emphasized to keep the manual intervention for data processing to a minimum to be scalable with area size and a readily available for operational use. The results for depth and volume, compared to a co-incidental ASO lidar-based measurements, showed almost identical statistics for mean and median for depth, and a similar estimation in volume for open areas at 1 m, 3 m, with 50 m showing the largest difference. The coinciding areas also had close values in SWE at the all resolutions. As with previous studies, vegetated, steep, or shallow snowpack areas had high reconstruction errors, with no measured snow depth and causing most of the missing areas in SCA. These terrain and snow depth characteristics accounted most for the missed volume by SfM compared to the ASO snow depth product.

We would like to see Structure from Motion applied to larger areas and more frequent image acquisition to improve our understanding of this technology at scale. As with lidar, it can provide high resolution spatially complete data sets with sub-meter accuracy. This capability can further improve our ability to model and understand snow-driven hydrological processes and contribute to explaining the consequences of our changing environment.

### Author contributions

JM and MS conceptualized the overall study, with helpful contributions from JD and DS on parts. JM performed the data processing and analysis. KB provided ASO data and support. MS provided financial support for the study. JM wrote the first draft of the manuscript, which was then contributed to by all authors.

### Code availability

The software components used to process data are publicly available on https://github.com/UofU-Cryosphere/snow-aso and analytical code for the output data is available on: https://github.com/jomey/raster_compare

### Data availability

Camera images, lidar point cloud files, and imaging spectrometer ground classification were acquired through personal communication with the ASO team. The snow depth map and values used for SWE are publicly available through the National Snow and Ice Data Center Distributed Active Archive Center download portal (Painter, 2018a, b, c).



**Competing interests**

Co-authors Jeffrey Deems and Kat Bormann were members of the NASA ASO team (which produced the data used in this
study). Jeffrey Deems is a co-founder of Airborne Snow Observatories, Inc. and Kat Bormann is currently employed by ASO,
Inc., formed as a result of the ASO NASA technology transition effort.

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
