# Peer review of "Mapping snow depth and volume at the alpine watershed scale from aerial imagery using Structure from Motion"

_Hydrology and Earth System Sciences, 2021_

## Author Comment (AC1)

*Hydrol. Earth Syst. Sci. Discuss., referee comment RC1 https://doi.org/10.5194/hess-2021-281-RC1, 2021 © Author(s) 2021. This work is distributed under the Creative Commons Attribution 4.0 License.*

[Figure]

**Comment on hess-2021-281**

*Anonymous Referee #1*
* * *
*Referee comment on "Mapping snow depth and volume at the alpine watershed scale from aerial imagery using Structure from Motion" by Joachim Meyer et al., Hydrol. Earth Syst. Sci. Discuss., https://doi.org/10.5194/hess-2021-281-RC1, 2021*
* * *
*Meyer et al. present an assessment of Structure from Motion (SfM) to map snow depth at basin scale. They do so by presenting two flights, where phtogrammetric images were captured in the context of the Airborne Snow Observaotry (ASO) lidar scans. This allowed authors to compare SfM to lidar scan. Results show that where snow was mapped by both ASO and SfM, the depths compared well, with a mean difference between -0.02 m and 0.03 m, NMAD of 0.22 m, and close snow volume agreement (+/- 5%). Limitations were found in vegetated areas, locations with shallow snow, and steep terrains. Overall, ASO mapped a larger snow area relative to SfM, with SfM missing ~14% of total snow volume as a result.*

*I enjoyed reading this manuscript, which focuses on an important topic: measuring high-resolution snapshots of snow depth at watershed scale using remote sensing.*

*Thank you, we appreciate you taking the time to read the manuscript and provide thoughtful feedback. The comments have improved the manuscript, and specifics about how we made updates in the manuscript to address the concerns are provided below (in blue).*

*Comparatively new techniques have emerged over the course of the most recent decades, including lidar, drones, and in fact photogrammetric flights. From this standpoint, the topic covered by this ms is certainly relevant and in line with the scope of HESS. At the same time, the comparatively large body of literature on these techniques (which the authors present in their ms) means that the novelty provided by this specific study is quite unclear. Some novelty points are highlighted at lines 55ff page 3, but they appear incremental to me. Also, the conclusion that SfM may be biased in areas with vegetation or shallow snow is not new. A more effective case should be made to justify publication.*

Secondly, results by these surveys look a little unconvincing with regard to SfM applicability, to the extent that the main conclusion of this manuscript (capturing large scale snow depth and volume with airborne images and photogrammetry could be an additional viable resource for understanding and monitoring snow water resources in certain environments) may be not supported by results. SfM missed about 14% of total snow, while snow volume was 86% of ASO volume. Fig 5 also shows clear biases in case of shallow snow cover, which overall leads to SfM underestimating snow depth (line 199).

From a presentation standpoint, the ms reads a little like a technical report. According to their aims and scope, "HESS encourages and supports fundamental and applied research that advances the understanding of hydrological systems, their role in providing water for ecosystems and society, and the role of the water cycle in the functioning of the Earth system." What are the specific research questions of the study that could justify publication in an international, broad journal? In other words, how could this survey be used to advance understanding of hydrological systems?

*To address the reviewer's concerns, we updated the abstract and the introduction, explicitly expanding on the value of this study relative to previous work on photogrammetric snow depth retrieval from piloted aircraft. Additionally, we added a section to the conclusions describing how this work is relevant given NASA's recent focus on incubating measurement methods (including photogrammetry) for Surface Topography and Vegetation (SVT), which includes snow depth, a targeted observable in the most recent Decadal Survey.*

*These updates justify the novelty of this study, better highlight where SfM could add value for expanded mapping of snow depth in the mountains, and support the broader relevance of photogrammetric snow depth mapping. The paper still provides the technical description that has interested many readers, supported by the >400 article views and downloads as a preprint, and added broader context and motivation for the work that makes it suitable for publication in HESS.*

*Included below are relevant updates to the text:*

*(Abstract)*

*Time series mapping of snow volume in the mountains at global scales and at resolutions needed for water resource management is an unsolved challenge to date. Snow depth mapping by differencing surface elevations from airborne lidar is a mature measurement approach filling the observation gap operationally in a few regions, primarily in mountain headwaters in the Western United States. The same concept for snow depth retrieval from stereo- or multi-view photogrammetry has been demonstrated, but these previous studies had limited ability to determine the uncertainties of photogrammetric snow depth at the basin scale. For example, assessments used non-coincident or discrete points for reference, masked out vegetation, or compared a subset of the fully snow-covered study domain. Here, using a unique data set with simultaneously collected airborne data, we compare snow depth mapped from multi-view Structure from Motion (SfM) photogrammetry to that mapped by lidar at multiple resolutions over an entire mountain basin (300 km2). After excluding reconstruction errors (negative depths), SfM had lower snow extent (~15%) and snow volume (~14%) compared to lidar. The reconstruction errors were primarily in areas with vegetation, shallow snow (< 1 m), and steep slopes (> 50o). Across the remaining snow extent, snow depths compared well to lidar with similar mean values (< 0.03 m difference) and snow volume (+/- 5%) across output*

resolutions of 1 m, 3 m, and 50 m, and with a normalized median absolute deviation (NMAD) of 0.22 m. Our results indicate that photogrammetry from aerial images can be applied in the mountains but would perform best for deeper snowpacks above tree line.

*(Introduction)*

Previous studies have demonstrated photogrammetric snow depth retrievals from piloted aircraft (Bühler et al., 2015; Nolan et al., 2015; Eberhard et al., 2020), but gaps remain for understanding uncertainties at the mountain basin scale. For example, Nolan et al., 2015 had a relatively flat terrain and small study typically not found in watersheds. Bühler et al., 2015 and Eberhard et al., 2021 used larger areas with representative alpine terrain, but only a smaller subset was compared to reference data. Additionally, both of these studies did not record the reference data simultaneously with images used for photogrammetry. This has important implications as the snowpack undergoes constant changes, and depth is unlikely to remain constant. From a methodology perspective, Bühler et al., 2015 also excluded all areas with visible vegetation in the snow scenes and did not analyze the characteristics of negative snow depths in the results. Additionally, the approach was different; the images were from a multispectral line scanner, and the snow depth map was produced in chunks. In contrast, Eberhard et al., 2021 had imagery from an RGB camera and reconstructed the study area as a whole. However, the processing step required manual placement of ground control points (GCP) for the scene to be geo-referenced accurately. The use of GCP makes it challenging to follow this approach in vast, remote, snow-covered areas. Their approach also aligned the snow depth via cubic-convolution resampling and left no understanding of the individual scenes' geo-location accuracy (snow-free and snow-on).

This work evaluates the ability of SfM to map snow depth distributions over a watershed with high-resolution RGB imagery captured by a piloted aircraft and its accuracy relative to a simultaneously collected lidar-based retrieval. The comparison is spatially complete, across different output product resolutions, with identical snowpack conditions, sensor viewing geometry, and environmental influences, such as the weather. The same data processing for snow-free and snow-on scenes reduces potential errors that could arise from combining different methods and datasets. The data sets are from an operational snow water resource mapping company, which could readily adapt this presented workflow. Meyer & Skiles (2019) showed that SfM can generate accurate snow surface DEMs from imagery collected from piloted aircraft over bright, freshly fallen snow. The compared snow-on surfaces from SfM and lidar had a relative accuracy of 0.17 m at 1m resolution. Building upon this work, we show in this paper that two SfM DEMs (snow-on and snow-free) can be used to calculate snow depths and corresponding snow volume over a larger alpine watershed (300 km2) across different output resolutions. The broader application of SfM will expand our understanding of the strengths and weaknesses of photogrammetric-based snow depth in the mountains, and ideally support broader use of aerial snow observations in terrain where these observations are suitable.

*(Methods)*

For consistency, the SWE calculation used the mean snow density calculated from the ASO SWE product distributed through NSIDC at the 50 m resolution (Painter, 2018c). SWE is a highly desired quantity for water resource forecasting and is commonly expressed in meters. In this study, we aggregated SWE as a sum of all pixels of measured snow depths and showed how depth differences propagate.

*(Conclusion)*

*We would like to see photogrammetry, including SfM, applied to larger areas and more frequent image acquisition to improve our understanding of this technology at scale. Our results show that it can provide spatially complete data sets with sub-meter accuracy across multiple output resolutions in the mountains and is best applied above the tree line and for deeper snowpack. This capability, possibly as a complement to lidar, can further improve our ability to monitor and understand snow-driven hydrological processes and environments. The importance of monitoring the mountain snow water reservoir is well recognized, with seasonal snow depth and snow water equivalent both being identified as 'targeted observables' in the most recent Earth Science Decadal Survey (National Academies of Sciences, Engineering, and Medicine, 2018). This United States National Academy of Sciences survey guides upcoming scientific missions and goals for earth observations from space. Targeted observables are priority observations that may not yet have mature measurement techniques but could within the next 10+ years. This recognition brings attention to emerging technologies and incubation funding to mature their approaches and application. Surface Topography and Vegetation (STV) is one such incubation effort, which focuses on high-resolution global topography mapping and topography change. Photogrammetry, along with lidar and radar, was specifically targeted as a measurement technology with potential for maturation (Donnellan et al., 2021). Although the focus is on stereo-photogrammetry, for which satellite capability is well established, the potential for spaceborne multi-view photogrammetry is also promising and equally suitable. Comparisons and data like those in this work contribute to the STV effort with methods undergoing active development. Ultimately, the goal is for global satellite-based time-series mapping of snow volume in the mountains and at resolutions needed for water resource management.*

*National Academies of Sciences, Engineering, and Medicine: Thriving on Our Changing Planet: A Decadal Strategy for Earth Observation from Space, The National Academies Press, Washington, DC, 716 pp., https://doi.org/10.17226/24938, 2018.*

*Donnellan, A., D. Harding, P. Lundgren, K. Wessels, A. Gardner, M. Simard, C. Parrish, C. Jones, Y. Lou, J. Stoker, K.J. Ranson, B. Osmanoglu, M. Lavalle, S. Luthcke, S. Saatchi, R. Treuhaft, Observing Earth's Changing Surface Topography and Vegetation Structure: A Framework for the Decade, NASA Surface Topography and Vegetation Incubation Study, 210 pp., 2021*

*I encourage authors to work on the above points, since obtaining snapshots of snow depth at basin scale is indeed a clear and important open issue in snow hydrology. I am looking forward to reading a revised version.*

*SPECIFIC COMMENTS*

*Line 47: various regulations exist at national and international level, which may limit the use of RPAS (e.g., over populated areas). This fact may be worth mentioning here.*

*We added this aspect to the paper.*

*RPAS have additionally limited areal coverage due to battery life, cannot be operated safely outside the line of sight, or face strong regulations in public areas*

*Line 134: why was the 1 m raster down sampled from the 3 m one, instead of being directly derived from the point cloud? What is the associated uncertainty?*

*It is indeed technically possible to export the point clouds from lidar at the higher resolution. However, the ASO workflow involves more steps than gridding the point cloud to produce the distributed snow depth map. This study focuses on the possibilities with SfM compared to the publicly available products at 3 m and 50 m. Downsampling to the higher resolution was done to demonstrate additional capabilities. The section that describes the creation of the 1 m ASO snow depth map has been revised:*

*This iteration resampled the 3 m ASO snow depths, kept the identical bounding box, and used the nearest-neighbor algorithm.*

*Line 196: 3770 m is unclear to me. Do you mean 3770m^3 (likely to small for watershed SWE) or 3770 mm on average?*

*This is the total amount of SWE when adding all the pixel values. We also added a description in the method section that explains why this number is presented as a 'm' value:*

*SWE is the desired quantity for water resource forecasting and is commonly expressed in meters. In this study, we aggregated SWE as a sum of all pixels of measured snow depths and showed how depth differences propagate*

*Line 285: despite being supported by some references, this threshold on 25 m for hydrologic models looks a little arbitrary. Hyper-resolution models are on the rise, also supported by satellite products that now exceed that threshold (e.g., Sentinel-2 images at 20 m).*

*We reworded the sentence to:*

*Inputs for hydrologic models, for instance, currently do not require or have been assessed against image resolutions of less than 25 m (Behrangi et al., 2018; Hedrick et al., 2020; Pflug and Lundquist, 2020)*

---

## Author Comment (AC2)

Hydrol. Earth Syst. Sci. Discuss., referee comment RC2 https://doi.org/10.5194/hess-2021-281-RC2, 2021 © Author(s) 2021. This work is distributed under the Creative Commons Attribution 4.0 License.

[Figure]

Comment on hess-2021-281

Anonymous Referee #2
* * *
Referee comment on "Mapping snow depth and volume at the alpine watershed scale from aerial imagery using Structure from Motion" by Joachim Meyer et al., Hydrol. Earth Syst. Sci. Discuss., https://doi.org/10.5194/hess-2021-281-RC2, 2021
* * *
Comments on "Mapping snow depth and volume at the alpine watershed scale from aerial imagery using Structure from Motion" HESS-2021-281

Meyer et al. present an evaluation of a snow depth map calculated from airborne visible images with the Structure from Motion (SfM) method. The snow depth map is compared with a synchronous snow depth map calculated from airborne LiDAR, considered as a reference. The authors suggest that the SfM snow depth map is suited for future large scale campaign as they find a little bias (<0.1 m) and a satisfying accuracy (NMAD<0.2 m) over 300 km² of mountain terrain partially covered with snow.

It is an interesting work which completes previous studies presenting new methods to map snow depth at high spatial resolution in mountain terrain using unmanned vehicle, airplanes or satellites combined with lidar or photogrammetry. The authors have a unique and rich pair of datasets at hand and they have made efforts to extract relevant information for the snow science community.

However, many points in the article need improvement. I listed below six major points which should be addressed. I know the amount of work which is implied by these remarks, but I believe it is necessary to make this article suitable for publication in HESS.

*We appreciate the reviewer taking the time to read the manuscript and provide thoughtful feedback. The comments have improved the manuscript, and specifics about how we made updates in the manuscript to address the concerns are provided below (in blue).*

**Major comments**

**Data and methods**

1.  The classification used is not clearly described. What date? What method is used? What
    are the different classes? It becomes very hard to understand l.144 : "Finally, the snow
    volume was compared, grouped by different surface classifications (snow, rock,
    vegetation; Figure 2a)"

It sounds like there is snow on pixels classified as rock and vegetation. But what is the
category "snow" then? In the same way, Figure 7b is hard to understand: we are looking
at snow depth on vegetation, rock and...snow?

*The land surface classification is produced from the ASO imaging spectrometer spectral reflectance,
each pixel is classified as one of four basic land surface classes (I.e it is not spectral unmixing). To
clarify this, we have added this to the data section:*

*Other data sets, acquired from ASO directly, were a land surface classification raster for the snow on
flight and the lidar point cloud from the snow-on and snow-free acquisition. The land surface
classification map is a standard output from the ASO imaging spectrometer spectral reflectance
processing pipeline, which categorizes each 3 m pixel into the basic land surface types; snow, rock,
vegetation, and water (Painter et al., 2016).*

*—-*

*The lidar can map snow even where the imaging spectrometer does not classify snow, for example in
the trees, which is a strength of lidar. For our purposes, snow extent is defined by where the lidar maps
snow depth. Still, the land surface classification is helpful to interpret results from SfM. The figure 7b
shows on which surface classification pixel type lidar did have a positive snow depth, but SfM did not
retrieve a positive depth value. (Figure 7 also confirms the conclusion that shallow snow is a
challenging SfM environment, as many values had less than 1 m in the ASO snow depth map.)*

Related to the classification: how are calculated the snow cover area (SCA) map? Is there
an independent SCA from SfM data and one from lidar data? Otherwise, how is there a
difference between both SCA (l.182)?

*The SCA for SfM is calculated relative to that from lidar, the reference. To clarify this, we have added
the methodology on the SfM SCA to the comparison section:*

*---*

*SCA for SfM was calculated as a percentage amount with pixels in the snow depth map that had a
positive value and relative to the total number of pixels with depth in the ASO map*

*---*

*The calculation of SfM SCA did not use the surface classification map and is a relative percentage to
the ASO depth map.*

What is the interest for this study to calculate the SWE? Here, this work focuses on mapping snow depth. Showing that SfM snow depths match lidar measurements is enough to show that valuable SWE maps can be further derived. Plus, as long as it is not clear how the SCA maps are calculated, the difference between SfM SWE and lidar SWE is hard to interpret. Using a single density factor and comparing it to a complex spatialized model could be the topic of another entire study.

*SWE is an important variable for hydrological water forecasting and relevant information for a broader audience that is more used to thinking in terms of water equivalent volume. For this audience, the SWE statistic shows how the missing snow depth propagates. It further shows that SfM could be well suited for forecasting applications in areas that closely match lidar (primarily open areas).*

*We added a section to clarify this in the methods:*

*SWE is a highly desired quantity for water resource forecasting and is commonly expressed in meters. In this study, we aggregated SWE as a sum of all pixels of measured snow depths and showed how depth differences propagate.*

**Results**

2. One main result is that there is no bias between SfM and lidar snow depth (Table 2,l.180). However, this is not what is suggested by Figure 5.a which is commented l.199 : "As a whole, SfM showed an underestimation of snow depth compared to ASO, using the 1 m resolution pixel-by-pixel values (Figure 5a)."

In this figure, one can see that the heat map is not centered at all on the one-to-one line. It is divided into two populations : one large (presumably from what can be inferred of the color scale) in which SfM snow depths are inferior to lidar and one in which lidar snow depth are between 0 m and 1 m and SfM snow depth are greater. Thus, it seems inaccurate to conclude that there is no bias in SfM snow depth. Besides, the latter population (where ASO snow depth is between 0 m and 1 m) needs more explanations. Is it related to errors in the classification? To errors in the lidar/SfM snow-on/snow-free DEM? To the downsampling of the ASO map?

*We agree that the heat map shown in Figure 5a shows inferior depths of SfM compared to Lidar. However, the conclusions that SfM has no bias stems from more than this figure. Figure 5a is a pixel by pixel comparison and our interpretation is that SfM does not match the depths with ASO in all locations. The slight shift signifies the concluded underestimation when compared to location. The big picture for the measured snow depths in the entire basin still compares well. For applications in water resource forecasting, the exact location is secondary to the total amount of snow.  Figure 5b shows a very good distribution agreement between the two techniques and the sample SWE*

*calculation results using coinciding measured snow depths confirms that too. With all the aspects combined, we concluded that there is no strong bias in SfM.*

*We updated the figure caption to:*

*SfM tended to underestimate the snow depth compared to ASO at the same pixel locations.*

3. Another main result is that SfM snow depths are less accurate for shallow snowpack. Is it accuracy relative to the snow depth? I cannot really imagine a reason for absolute accuracy to be worse for shallow snowpack. The surface of a shallow snowpack should appear just the same as the surface of a deep snowpack on an airborne image. Please give us your opinion on this point.

*The lower accuracy for a shallow snowpack is cause by two aspects in this work:*

   1. *With the NMAD of 0.22m, any snow depth below that value is less likely to be reconstructed successfully. This is also shown in the histogram in Figure 5b, where the depths peak shortly after the 0 m. ASO on the other hand, has its peak at 0 m, which could be the more realistic representation. With this flight being late in the melt season (25th of May), there should be a lot more areas with little to no snow.*

   2. *Shallow snow in open areas is also influenced by the snow-free (summer) ground cover. Tall grasses or bushes for instance are a difficult ground cover for SfM to reconstruct precisely. This high potential source of error propagates when using the snow-free DEM for the depth calculation.*

*We expanded the result sections in 5.3 to add this perspective:*

*This showed that SfM failed to reconstruct where ASO mapped shallow snow (<1 m; Figure 7a) in open areas. The largest negative SfM values (-5 m and -28 m) were primarily found in areas classified as vegetation (Figure 7b). The findings with shallow snow-depth were expected and any depth below the NMAD of 0.22 m is less likely to be retrieved successfully. Shallow vegetation in the snow-free scene is additionally compacted through snow deposition in the winter and a hard to account for physical process with the differencing principle. This is especially true when SfM cannot reconstruct land surface covers like grassland or shallow bushes in the snow-free scene well, whereas lidar can map more accurately. Overall, these results are in line with previous SfM work, where shallow depth or forested areas showed to impair the ability of SfM to measure accurate surface elevations (Avanzi et al., 2018; Fernandes et al., 2018).*

Comparison to existing studies and novelty of the work

4. The article is not strongly embedded in the existing literature. The need for more evaluation of SfM snow depth is justified by a single sentence in the introduction (l.56-l.59). This seems a bit short as three previous studies calculated snow depth map from

airborne SfM. These studies are marginally used in the discussion. Bühler et al. (2015) is not even used any further in the article, although they calculated a 145 km² snow depth map with airborne SfM. The authors should clarify what is the added value of their study with respect to the current knowledge.

*We have added additional content to the abstract and the introduction to highlight the gap filled by this work, and address the reviewer's concern:*

*Abstract*

*Snow depth mapping by differencing surface elevations from airborne lidar is a mature measurement approach filling the observation gap operationally in a few regions, primarily in mountain headwaters in the Western United States. The same concept for snow depth retrieval from stereo- or multi-view photogrammetry has been demonstrated, but these previous studies had limited ability to determine the uncertainties of photogrammetric snow depth at the basin scale. For example, assessments used non-coincident or discrete points for reference, masked out vegetation, or compared a subset of the fully snow-covered study domain. Here, using a unique data set with simultaneously collected airborne data, we compare snow depth mapped from multi-view Structure from Motion (SfM) photogrammetry to that mapped by lidar at multiple resolutions over an entire mountain basin (300 km2).*

*Introduction*

*Previous studies have demonstrated photogrammetric snow depth retrievals from piloted aircraft (Bühler et al., 2015; Nolan et al., 2015; Eberhard et al., 2020), but gaps remain for understanding uncertainties at the mountain basin scale. For example, Nolan et al., 2015 had a relatively flat terrain and small study typically not found in watersheds. Bühler et al., 2015 and Eberhard et al., 2021 used larger areas with representative alpine terrain, but only a smaller subset was compared to reference data. Additionally, both of these studies did not record the reference data simultaneously with images used for photogrammetry. This has important implications as the snowpack undergoes constant changes, and depth is unlikely to remain constant. From a methodology perspective, Bühler et al., 2015 also excluded all areas with visible vegetation in the snow scenes and did not analyze the characteristics of negative snow depths in the results. Additionally, the approach was different; the images were from a multispectral line scanner, and the snow depth map was produced in chunks. In contrast, Eberhard et al., 2021 had imagery from an RGB camera and reconstructed the study area as a whole. However, the processing step required manual placement of ground control points (GCP) for the scene to be geo-referenced accurately. The use of GCP makes it challenging to follow this approach in vast, remote, snow-covered areas. Their approach also aligned the snow depth via cubic-convolution resampling and left no understanding of the individual scenes' geo-location accuracy (snow-free and snow-on).*

Also, the conclusions and finding of this article (Meyer et al., 2021) should be compared with the ones from Meyer and Skiles (2019). The main innovations of this article (2021) are that a snow-free DSM is used and have to be correctly geo-located before differencing snow-on and snow-off DSM. What was learned from that? Is the accuracy measured in

this article in line with the accuracy measured in Meyer and Skiles (2019)? Since a snowfree DSM is used : are there larger errors in the snow-on or snow-free DSM?

*The revisited introduction section expanded on the differences between this and the preceding work of Meyer & Skiles 2019. The latter compared two snow surfaces, aligned to each other across their full extent, before assessing the SfM quality. The resulting NMAD there cannot be directly compared to the NMAD in this work, as here it is retrieved through only using control surfaces. The quality of the individual snow-on and snow-free DEMs is discussed in section 6.3 through the result of the determined shift vectors from the co-registration.*

*Below the revised introduction section:*

*Meyer & Skiles (2019) showed that SfM can generate accurate snow surface DEMs from imagery collected from piloted aircraft over bright, freshly fallen snow. The compared snow-on surfaces from SfM and lidar had a relative accuracy of 0.17 m at 1m resolution. Building upon this work, we show in this paper that two SfM DEMs (snow-on and snow-free) can be used across different output resolutions to calculate snow depths and corresponding snow volume over a larger alpine watershed (300 km$^2$). The broader application of SfM will expand our understanding of the strengths and weaknesses of photogrammetric-based techniques and provide more areal snow observations.*

**Language**

6. Many sentences do not read easily. Some are too long. It often brings confusion. I am not a native english speak myself but I feel that the manuscript should
be carefully revised before next submission. For examples, see specific comments on l.22, l.60, l.104, l.118, l.142-143, caption of Figure 4…

*We responded to each of the highlighted line comments in the section below.*

**Specific comments**

l.20 : "certain environment" Please precise.

*This sentence has been updated through revisions.*

l.22 : "Snow depth and SWE" are "applied"?
Updated to:

*Snow depth and snow water equivalent are essential monitored quantities and utilized in many water resource applications*

l.27 : remove "differentially"

*The paragraph has been and this section reworded to:*

*The gap can be addressed with remotely sensed surface elevation products by estimating snow depth through subtracting snow-free elevations from snow-on elevations (Deems et al 2013).*

l.31 : "which resulted in varying degrees for monitoring snow depth accuracy" what does it mean?

*Revisited the sentence and named the difference more concretely:*

*This principle has been demonstrated from several remote sensing platforms, spanning a range of spatial resolution, areal coverage and repeat intervals and had varying accuracy for snow depth*

l.32 : This article, as stated l.28, focuses on raster-based products. Treichler and Kääb did not use rasters. I suggest removing its citation.

*We changed the preceding paragraph and explained the principle more broadly (response to comment on l.27). This allows for a comparison of the differencing principle to more datasets.*

l.35 : To my knowledge, Marti et al. (2016) was the first study to calculate snow depth from photogrammetry satellite. To my understanding, no snow depth is calculated in Shean et al. (2016). They rather discuss how to "limit the influence of points acquired when seasonal snow was present on bedrock surfaces". Consider swapping both references.

*References updated.*

l.44 : "up to alpine catchment size" do alpine catchment have a typical upper area boundary? A quick google scholar search of "alpine catchment" provides article studying alpine catchment of more than 1 km².

*The area size was removed and no constraint to alpine catchment size is stated.*

l.49 : Avoid "/", maybe instead : "high resolution and high accuracy"

*Updated*

l.49 : "over any desired target area" This is arguable. Some facilities are needed nearby.

*It is true that airborne campaigns require infrastructure close by. However, we don't see this as a limiting factor, as many areas of interest with seasonal snow coverage are accessible via airplane.*

l.59 : "the coincidental collection with this study"?
l.60 : "the coincidental collection […] as well as precluding measurement errors due to manual recording such as snow probes" shorten this sentence (It enables precluding => it precludes), and check the meaning : it is not the coincidental collection which precludes errors in snow probes measurements.
l.61 : "the data **used**"

l.62 : "Meyer & Skiles (2019) showed that accurate DEMs can be generated from imagery collected from piloted aircraft over bright snow surfaces using SfM" It sounds a bit like this is first time it was done, in contradiction with l.56. Rewrite this paragraph to enhance the novelty of this work.

*The paragraph from line 54 to 69 was revised and addresses the style and wording comments. The differences between this and previous work is now explained more explicitly.*

l.71 : "E E"

*Corrected.*

l.99 : Is there a reason for this difference in acquisition interval?

*The imagery is currently not used by ASO post flight, but rather by flight operator's during flight to monitor conditions beneath the plane. Hence, the interval can vary depending on the flight operator's needs. This explanation was added to the text as well.*

l.102 : It sounds like only an ASO snow depth map was used. At least the ASO snow-on DEM and the classification seemed to have been used. Include them clearly in the Data section.

*Added additional information to the data section and listed the land surface classification along with the snow-on lidar point cloud.*

l.103 : "the identical difference principle" what is this? If it is not a usual term, please delete or rephrase.

l.104 : "subtracted with" reformulate : subtracted from

*Revisited the sentence to address both comments.*

l.117 : consider rephrasing with a more direct structure. Maybe something like : "Controle surfaces of the ASO snow-on acquisition flight was used as a reference."

*The section of co-registration was revisited to improve clarity of the process and steps taken (addressing this comment, and the three below).*

l.117 : "ASO snow-on acquisition flight" : it is not the flight which is used. Is it a pointcloud? The 3 m DSM? Please explain in the data section.

l.118 : "have" instead of "are"?

l.118 : "identified" how? See comments on classifications.

l.122 : "An added advantage of co-registering of the SfM point clouds" delete "of"

*The section of co-registration was revisited to improve clarity of the process and steps taken.*

l.133 : "3 **m** and 50 m."

*Updated*

l.135 : "with SfM supporting the higher resolution through the previously mentioned high point density," not necessary, long sentence, grammatically peculiar.

*Changed to:*

*We additionally compared the products at the 1 m resolution with SfM supporting the resolution through the high point density.*

l.136 : "downsampled" : How? Which method? Can we estimate what snow depth accuracy can be reached after this operation? This is important. It is not advised to downsample any raster data especially since there is a high-density point cloud that could be rasterized at 1 m.

*Reworded the section that describes the comparison at 1 m.*

*While a native ASO snow depth product at the 1 m resolution is indeed possible, there are more steps in the ASO workflow than gridding the point cloud to produce the distributed snow depth map. This study focuses on the possibilities with SfM compared to a publicly available product. Producing a snow depth map based on a lidar point cloud, and accounting for the challenges with that process is out of the scope for this work.*

*The revised description for the "downsampled" section:*

*This iteration resampled the 3 m ASO snow depths, kept the identical bounding box, and used the nearest-neighbor algorithm.*

l.142-143 : "The analysis[…] binned the depth by elevation to assess similarities in the vertical relief" I don't understand.

*The elevation band comparison is intended to assess any possible challenges for SfM as elevation increases. For instance, does SfM underperform in lower elevations with more vegetation? Do higher elevations with steeper cliffs present more difficult terrain? These two questions are addressed with Figure 5 and section 5.2.2.*

l.155 : "stable terrain"? "control surface"?

*Corrected*

l.163 : "the raster products exported from the respective SfM point clouds" long sentence. Maybe replace with "the DSM rasterized from ..."

*Reworded*

l.164 : "the remaining difference in the NMAD" : not clear. Difference between NMAD and SD?

*Combined this and the above comment to rework the paragraph to:*

*After applying the translation to the respective SfM point clouds, the control surfaces in the SfM DEMs had a mean of 0.02 m with a standard deviation of 0.52 m, and median of 0.03 m. The NMAD of 0.22 m indicated that there were still some outliers in the control surfaces, despite all the refinements to constrain those.*

l.165 : A bad co-registration could occur and still provide a zero median/mean. Compare rather NMAD before and after co-registration.

*We see limited value in such a comparison. The snow-on lidar point cloud and land surface classification are co-registered to each other. Using the classification on the unregistered SfM clouds will include incorrect surface types. Hence a comparison of the before and after numbers would not be over the same surfaces. The given shift vector additionally supports this with corrections in the North and East.*

l.167 : This is often done but could be discussed in this article. If the snow-on and snowoff lidar DSM are available to the authors, the difference of both should be calculated, producing a difference of lidar DEM (DoD). This would be immensely interesting to provide the NMAD of the control surface of this DoD and compare it to the NMAD of the SfM products. Please consider providing these numbers.

*We agree that this is an alternate approach to determine the uncertainty in ASO products. We do not calculate this number, related to that fact that we don't produce our own snow depths maps from the lidar point cloud (our response to l.136). An export of a DEM with control surfaces has to be identical to the DEMs that are used to calculate the ASO snow depth map, which is not within the scope of this work. The ASO team communicates an uncertainty for their published maps, and here our intent is to compare SfM snow depth map to their operational snow depth product.*

l.169 : Long, convoluted sentence. I would suggest something around : "Overall, there was good agreement in both snow depth and snow volume where snow depth was measured in both the SfM and ASO depth maps (Figure 3)." Introduce "SfM" and "ASO" somewhere in the methods.
l.175 : missing "and"?

*Both comments addressed by rephrasing the paragraph:*

*Overall, where SfM and ASO measured snow depth, there was a good agreement between snow depth and snow volume (Figure 3).*

l.177 : do we need "in the distributed product through NSIDC" Clarify this in the Data&Methods so that it does not show up in the results.

*Removed the phrase.*

l.179 : maybe remove capital letters of Mean Median Standard Deviation?

*Agreed. Removed capitalization for consistency with the text.*

l.183 : "The SfM SCA coincided with ASO at the 1 m resolution by 72%, 73% at 3 m, and 64% at 50 m, showing a small difference between the lower and higher resolution" Please rephrase : "to coincide by N%" is not a clear formulation to compare two simple figures.

*Reworded to:*

*The SfM SCA matched the ASO SCA at the 1 m resolution by 72%, 73% at 3 m, and 64% at 50 m, showing a small difference between the lower and higher resolution.*

l.192 : "a similar range of **map** resolution"?

*Changed to "output resolutions'*

l.194 : median (here) or mean (l.141) density?

*Corrected to mean.*

l.194 : "less variable" this should not be in results but in discussion.

*The discussion does not go into the aspects of using this simplification for density. This added information sets up the justification that the fixed value is a well-suited approach and matches reasonably with the official SWE map.*

l.196 : "m" of SWE sounds odd for total basin SWE. Is it a common term?

*In situ snow monitoring infrastructure in the United States (SNOTEL sites) commonly report the observed SWE as mm. Our comparison aggregates the SWE over the entire basin and we converted the value to meters for interpretation. Meters are also used in other papers for basin integrated values, including in the ASO paper (Painter et al., 2016). The description of this approach was added to the comparison section (4.3):*

*SWE is a highly desired quantity for water resource forecasting and is commonly expressed in meters. In this study, we aggregated SWE as a sum of all pixels of measured snow depths and showed how depth differences propagate.*

l.199 : "As a whole, SfM showed an underestimation of snow depth compared to ASO, using the 1 m resolution pixel-by-pixel values (Figure 5a)." In Table 2 and in the paragraphs commenting it, ASO and SfM mean/median snow depth agree within a few cm. This seems contradictory. See main comment

*Please see our response to the main comment #3.*

l.201 : "highly localized"? small areas?

*Correct*

l.205 : "coinciding area"?

*Changed to:*

*at the coinciding area with snow depth*

l.205 : are we talking about the land occupation under the snow? See main comment.
l.205 : grammar.

*Additional information was provided for the land surface classification map. This sentence changed to:*

*A closer look at the coinciding area with snow depth at the 1 m resolution had most SfM pixels classified as open snow (81%), followed by rock (12%), and then vegetation (7%) land surface types.*

l.207 : "capturing 92 %" unclear.

*Changed to:*

*A closer look at the coinciding area with snow depth at the 1 m resolution had most SfM pixels classified as open snow (81%), followed by rock (12%), and then vegetation (7%) land surface types. In these types, SfM underestimated snow volume relative to ASO in the pixels classified as snow, matching 92%, while overestimating snow volume in the rock and vegetation pixels.*

l.208 : This is hard to grasp "For ASO, snow volume was distributed differently across land surface types in the entire watershed; 69% in open snow, 15% in rock, and 16% in vegetation." What is rock/vegetation? Rock/vegetation covered with snow? But what is open snow then?

*Please see the response to the main comment regards the surface classification types.*

l.209 : "in part" what else?

*Comparing the snow volume across the pixel classification was a first indication that SfM had less snow mapped within vegetation. The second part to this is shown in the comparison of all the negative SfM values, where there were a lot of high negative values (Figure 7a).*

l.212 : "start**ing"**

*This should remain 'started' (past tense) as it had increasing values that started to decrease in higher elevations.*

l.220 : treatment of negative snow depth and gaps should be explained in method.
5.3. Confusing section. I understand that "gaps" are defined as SfM without measurements and "missed area" as SfM with negative snow depth. Then l.220, "missed snow depth" have a positive (!) mean/median. And l.224-225 you mention negative snow depth value related to where SfM failed (!) to map snow.

*The treatment was moved to the methods section.*

*Clarified that the positive mean, median, and standard deviation was from checking the values in the ASO map. The paragraph was revisited and clarified that these values were obtained from the ASO depth map:*

*The ASO snow depth pixels that were missed by SfM had a mean of 0.48 m, median of 0.39 m, and a standard deviation of 0.42 m.*

l.248 : grammar?
l.252 "and the image RGB information (Shaw et al., 2020) or near-infrared spectrum (Deschamps-Berger et al., 2020)" These studies use satellite images. Here it sounds like they used airborne SfM.

*The paragraph was separated into its own section "Availability of reference data". With this we want to make clear that both citations are not airplane-based studies and is focused on the input and output data. The sentence was reworded to:*

*For land surface classification, the snow-on and snow-free products can be directly classified using the photogrammetric DEMs and the image RGB information (Shaw et al., 2020) or near-infrared spectrum (Deschamps-Berger et al., 2020), where available*

l.264 : what is "perspective information"?

*Updated to:*

*We believe that image geo-location and perspective information (omega, kappa, and phi) combined with co-registration is a reliable substitute to GCP's, while not compromising on output quality*

l.266 : the first application of "photogrammetric snow depth products from satellite image" was by Marti et al., (2016). It is missing in the list of cited work.

*This section focuses on the use of co-registration for photogrammetric products. While Marti et al., 2016 also did that, the correction in the vertical was only performed with a single geo-location outside the study domain. The three cited studies here used multiple control surfaces within their domain and is more in line with our approach. Other important difference is that Shean et al., 2016 and McGrath et al, 2019 performed the co-registration via the same methodology in this study; a 3-D transformation determined with the ICP algorithm.*

l.269 : "We hypothesize that the snow-free scene, with more exposed vegetation and ground cover, degraded the accuracy for SfM." I understand that for vegetation. But why would ground cover degrade accuracy for SfM?

*Ground cover is referred to as different type of grass land for instance, which are another challenging surface type to reconstruct as it provides little surface area for SfM.*

l.272 : "it is further feasible to align the two models to each other and compute snow depth and volume in relative geo-location space." Cite the work who did that, otherwise it is an assumption.

*We used the word 'feasible' to indicate this as a hypothesis. Technically, there is no restriction in this workflow to align the two SfM point clouds to each other. To our knowledge, no such work has been performed.*

l.272 : "Alpine areas benefit from having exposed control surfaces for multi-view image processing and co-registration, having identifiable features in both scenes." I am not so sure about that. A counter-example would be an image acquisition after a fresh snowfall reaching the tree line.

*We generalized the environmental description that support geo-referencing to visible control surfaces.*

*Generally, alpine areas with seasonal snow cover benefit from having exposed control surfaces for multi-view image processing and co-registration and being identifiable features in both scenes.*

l.276 : "image" the influence of the resolution of the source image is not discussed here.It is rather the resolution of the snow depth map. Please rephrase.

*We changed the section title to "Influence of snow depth map resolution" and updated the wording accordingly.*

l.278 : "the overlapping pixel wise difference" : unclear.

*Updated to:*

*The pixel wise difference in overlapping areas*

l.280 : "0.05 m (ASO, personal communication)." there must be a publication justifying this number. Otherwise, use the two ASO DEMs to calculate the difference over control area.

*We changed the number to the last published (0.08 m) in Painter et al., 2016.*

l.287 : "50 m there" Where?

*Removed 'there'.*

l.293 : satellites do not really have a larger coverage area (300 km²), but : less images, less radiometric depth, slightly lower image resolution, less known attitude (jitter)...

*Updated the wording to:*

*Reasons for the higher accuracy are primarily technical, as satellite stereo pairs have, for instance, a larger area or lower resolution in a single image, making it more difficult to capture high enough terrain detail for reconstruction.*

l.298-299 : "atmospheric features like clouds" what other atmospheric features do you think of?

*Another example could be fog low down in the atmosphere or higher amounts of water vapor higher up that create hazy conditions, but have yet to form into a cloud.*

l.300 : "On the smaller scale using RPAS platforms, the accuracy is higher" sounds odd.

*Removed 'platforms' as it indeed sounded like an unnecessary addition to the acronym.*

l.314: could 6.5 be shortened and merged with the conclusions?

*We would like to keep this section separate from the conclusions as it recaptures the advances in hydrology due to the availability of spatially complete snow depth maps where SfM can contribute. It also serves as an outlook on where we see suggest future adaptions of SfM and how to improve the application independent of the presented findings.*

l.317: too convoluted.

*Shortened and revised:*

*Spatially complete and temporally extensive consistent records can further accelerate our ability to understand snow processes at scale and have also been shown to improve estimates of SWE (Margulis et al., 2019) and streamflow (Shaw et al., 2020) through assimilation.*

l.317 : Consider adding Brauchli et al. (2017)

Brauchli, T., Trujillo, E., Huwald, H., & Lehning, M. (2017). Influence of slope-scale snowmelt on catchment response simulated with the Alpine3D model. Water Resources Research, 53, 10,723–10,739. https://doi.org/10.1002/2017WR021278 l.319 : .) ).

*Added.*

l.320 : I am not sure of the benefit of the first lidar flight. Lidar and SfM products have similar coverage and accuracy. Pflug and Lundquist (2020) suggest combining much more different datasets like a continuous accurate lidar snow depth with repeated measurements on a small portion of the terrain (<4 % of total surface).

*Plfug and Lundquist suggest the use of multiple sources along with airborne lidar surveys in pattern repeatability studies (section 7.3). The idea here is to get the more accurate observation across all surface types (lidar) to start, then see if successive observations require the accuracy of lidar or if SfM can be sufficient over a domain subset. The sentence was revised to:*

*Although SfM does not yet deliver similar accuracies across all terrain characteristics and land cover classes, it can be used to supplement or build upon lidar data sets. For instance, a first survey could be conducted using the more accurate lidar, and successive observations use the more cost-efficient SfM for open areas with little forested areas as a subset of the domain (Pflug and Lundquist, 2020).*

l.322 : "we are encouraged that SfM can be an option" grammar.

*We reworded to:*

*The results of this work support that SfM can be an option for operations*

l.324 : "it is further feasible **to source the images**" is it a common formulation?

*Changed to:*

*it is further feasible to use the images*

l.324 : "from space-borne platforms" which satellites?

*Bhushan et al., 2021 showed the use of triplet and satellite videos from SkySat. One possible option could be to use the videos and produces multiple images. Then use SfM to reconstruct the area.*

*We added the citation to the main text:*

*From a technical setup perspective, it may be feasible to use images or video from space-borne platforms (Bhushan et al., 2021), adding the option of temporally consistent broad-scale coverage, and reducing operational requirements.*

l.327 : "at hand"?

*We revisited this sentence.*

l.330 : "It also **emphasized to keep** the manual intervention for data processing to a minimum to be scalable with area size **and a readily available for operational use**." grammar?

*Changed to:*

*It also emphasized keeping the manual intervention for data processing to a minimum to be scalable with area size and readily available for operational use.*

l.333: "at 1 m, 3 m, with 50 m showing the largest difference" grammar?

*Changed to:*

*resolutions at 1 m and 3 m, with the largest difference at 50 m.*

l.334: "at the all resolutions"

l.339 : "high resolution spatially complete data" too long.

l.341 : "contribute to explaining the consequences of our changing environment" too general.

*The conclusion section was revisited to address the comments from l.333 to l.339*

**Figures and tables**

Figure 4. I do not understand what you mean with "compensate". I guess that the amount of missing areas at 50 m depends on how the setup of the IDW algorithm, doesn't it? Plus, the end of the caption is not clear "Higher resolutions at 1 m and 3 m […] and successfully measured more areas with snow depths at edges of vegetation were more distinct." Scale is missing.

*Updated the caption to:*

*Comparison of snow depth across 1 m, 3 m, and 50 m resolution. SfM did not accurately capture snow depth within vegetated areas, but showed good agreement for open areas. Higher resolutions at 1 m and 3 m measured more areas of snow depth around vegetation and had closer values of SCA compared to ASO.*

Fig 5.a : There is a constant set of point ~0.2 m in ASO snow depth. Please comment on that.

*We believe this comment is for figure 5b as well. Please see our remarks there*

Fig 5.b : I agree that the ASO snow depth should be considered as the reference. But here, the SfM smooth shape of histogram looks more convincing than the discontinuous shape

of ASO's. Can you please comment on the discrepancy between ASO and SfM for snow depth<~0.6 m.

*The drop off in the SfM curve before reaching 0 m of depth confirms the conclusion that SfM does not measure shallow depths well. We also want to remind that the histogram is only showing the coinciding area between SfM and ASO. There are a lot of areas missing in SfM, where ASO measured depths of less than 1 m. With this, we revisited the caption to remind the reader that this figure is not using all measured depths by ASO.*

I guess the two "mean" lines perfectly overlap. Find a visual way to show it like using two colored dashed lines and shifting the vertical origin of one of them.

*The figure was revised and removed the lines for means and uncertainty to improve clarity. It also zoomed in for a) with the updated colorbar.*

[Figure]

Figure 6 : this figure could be more informative. Most of the colored area is pitch black. We cannot really tell how the two distributions compare. Consider changing the colorbar.

*The colorbar increased to a max pixel count of 5000 vs 1000 before. The same colorbar is also used and updated accordingly in Figure 5 for consistency and to assist interpretation.*

[Figure]

Figure 8 : "median SfM snow depth" : further in the caption, it sounds like only the negative SfM snow depth selected.

There are three orange dotted line, they cannot all show -1 m limit. It is hard to tell at this scale but it does not seem like a "linear trend" for slopes < 55°.

*We updated the figure and caption. The style for the helper line at 55 degrees is now different to better distinguish with the helper line for -1 m median snow depth difference. The 85-degree line was removed as it seemed not helpful after review.*

[Figure]

Table 1: Add the year to the dates.

*Added*

Table 2: Confusing what the column "Difference" is.

From the text (l.144) "calculated by subtracting the SDSfM from SDASO" => difference = SDASO-SDSFM

1st row of the table: difference = SDASO-SDSfM

All the mean and median row: difference (0.01) = SDSfM (1.06) – SDASO (1.05)

*The text describing the comparison were revisited and the table updated accordingly. The table now consistently shows the statistics for $SD_{SfM} - SD_{ASO}$*

Table 2: Include the note in the caption. What is "overlapping area by SfM"?

*Updated to:*

*Mean, Median, and Standard Deviation are for coinciding areas by SfM and ASO.*